

# Medical cannabis use in Thailand after its legalization: a respondent-driven sample survey

Sawitri Assanangkornchai[1], Kanittha Thaikla[2], Muhammadfahmee Talek[3] and Darika Saingam[1]

[1] Department of Epidemiology, Faculty of Medicine, Prince of Songkla University, Hat Yai, Songkhla, Thailand
[2] Research Institute for Health Sciences, Chiang Mai University, Muang, Chiang Mai, Thailand
[3] Faculty of Nursing, Prince of Songkla University, Pattani Campus, Muang, Pattani, Thailand

Corresponding author
Sawitri Assanangkornchai,
savitree.a@psu.ac.th

## ABSTRACT

**Background.** Many countries now allow the consumption of cannabis or cannabinoids for medical purposes with varying approaches concerning products allowed and the regulatory frameworks prevailing their endowment. On 18 February 2019 Thailand passed legislation allowing the use of cannabis for medical purposes. This study aimed to examine patterns and purposes for consumption of medical cannabis, and consumers' perceptions and opinions towards benefits and harms of cannabis and related policies in 2019–2020.

**Methods.** A cross-sectional study using a respondent-driven sampling (RDS) method was conducted in four sites across Thailand. Participants were 485 adults aged 18 years and over, living in the study region, who had used cannabis for medical purposes within the past 12 months. Face-to-face interviews using a structured questionnaire were used to collect data on (1) demographic characteristics, (2) pattern of consumption, (3) source of information and perception of benefits and harms of medical cannabis, and (4) opinion towards cannabis policies. Data were analyzed using RDS Analyst and presented as percentage and mean with 95% confidence interval (CI).

**Results.** Most participants (84.7%, 95% CI [78.9–90.5]) used an oral form of crude oil extract while 9.2% (95% CI [4.1–14.2]) used the raw form. The most common uses were for treatment of cancers (23.3%, 95% CI [16.1–30.4]), neuropsychiatric symptoms (22.8%, 95% CI [17.5–28.0]), and musculoskeletal pains (21.6%, 95% CI [16.7–26.6]). Illegal sources such as underground traders (54.5%, 95% CI [40.8–68.3]), friends and relatives (12.2%, 95% CI [6.2–18.3]), not-for-profit provider groups (5.2%, 95% CI [0.5–10.9]), and clandestine growers or producers (2.9%, 95% CI [0.6–5.3]) were the main suppliers. Most (>80%) perceived cannabis could treat cancers, chronic pains, insomnia, Parkinson's disease and generalized anxiety disorder. Less than half perceived that cannabis could cause adverse conditions *e.g.,* palpitation, panic, memory impairment and schizophrenic-like psychosis. Most respondents agreed or strongly agreed with the policies regarding permission to use cannabis for medical purposes (95.1%, 95% CI [92.0–98.2]), for the legal sale of medical cannabis products (95.9%, 95% CI [93.7–98.2]), and for people to grow cannabis for medical use (94.2%, 95% CI [91.8–96.5]). However, only two-thirds agreed with policies concerning the sales of cannabis (65.3%, 95% CI [56.9–73.7]) and home-grown cannabis for recreational purposes (61.3%, 95% CI [52.7–69.9]).

**Conclusion**. Our study reports the experiences of consumers of medical cannabis in the first year after its legalization in Thailand. Consumers reported various patterns and indications of consumption that were not supported by scientific evidence, but had positive perception of the results of consumption. These findings highlight ongoing policy challenges for Thailand and can be a lesson to be learned for other countries in the region.

## INTRODUCTION

The past two decades have seen a global trend towards the legalization of cannabis for medical purposes (*Aguilar et al., 2018*), reflecting increased evidence of its efficacy and patient interest in the use of cannabis and cannabinoids for treatment of several conditions (*National Academies Press for the National Academies of Sciences Engineering and Medicine, 2017*; *European Monitoring Centre for Drugs and Drugs Addiction, 2018*). Many countries, for example the USA, Canada, Israel, Argentina, Australia and most countries in Europe, now either allow or are considering allowing the consumption of cannabis or cannabinoids for medical purposes with varying approaches concerning type of products granted and the regulatory frameworks prevailing their endowment (*Aguilar et al., 2018*; *European Monitoring Centre for Drugs and Drugs Addiction, 2018*).

In Thailand, movement towards legalizing cannabis started in 2016 and gained momentum in 2018–2019 when an elected political party used it as a priority policy for the general election in March 2019. Medical cannabis was officially legalized in Thailand on February 18, 2019, making it the first country to do so in Southeast Asia. The "Narcotics Act of 2019" is a modification of the Narcotics Act of 1979, whereby cannabis was still classified as a class-5 narcotic and the recreational use of the drug remains illegal. Thai people are now allowed to apply for cannabis treatment of their medical condition(s). Research, cultivation and processing, and the import and export of cannabis are also conditionally permitted. Governmental and research organizations, medical practitioners, including doctors, dentists, pharmacists, veterinarians, traditional health practitioners and patients are granted licenses to either consume, possess, research, or produce and trade in cannabis according to particular guidelines (*Government of Thailand, 2019*).

After the enactment of this Act, and the general election, medical cannabis became a national agenda and a priority policy of the Ministry of Public Health (MoPH), with several interventions being implemented to promote its access and medical use. Three groups of medical conditions are included in the list of indications for medical cannabis treatment by the Ministry of Public Health (MoPH), namely (A) conditions with strong evidence of benefits from medical cannabis, *i.e.*, chemotherapy-induced nausea and vomiting, intractable epilepsy, spasticity in patients with multiple sclerosis and neuropathic pain, (B) conditions with some evidence of benefits, *i.e.*, patients in palliative care, patients with

end-stage cancer, Parkinson disease, Alzheimer disease, generalized anxiety disorder and other demyelinating diseases, and (C) conditions which may be benefited from treatment with cannabis should there be more evidence in the future, *e.g.*, cancers of some organs. Additionally, 16 regimens of the Thai traditional medicine were approved (*Ministry of Public Health, 2020*).

Several frameworks and guidelines have been developed for the jurisdiction and licensing for medical cannabis. In July 2019 the recommendations on cannabis treatment and care in Thailand were published by the MoPH, covering the use for a range of conditions of both modern and traditional medicines. Health professionals and Thai traditional doctors have been trained in a short training course and were granted a license to prescribe cannabis or cannabinoids (*Department of Medical Services , 2020*). Currently, 339 medical cannabis clinics and 449 Thai traditional medicine clinics in the MoPH hospitals have provided medical cannabis treatment (*Committee for Public Relations on Medical Cannabis of the Ministry of Public Health, 2021*). Three groups of medical cannabis products, including registered drugs per the new Narcotics Act, Thai traditional medicine having approved compositions (16 regimens), and folk-doctor cannabis oil have been approved for medical use (*Committee for Public Relations on Medical Cannabis of the Ministry of Public Health, 2021*). Licensed healthcare practitioners, including medical doctors, dentists, Thai traditional medicine doctors and folk healers can prescribe these products, registered under the Special Access Scheme (SAS), for patients to use for medical purposes.

This landmark change in policy has markedly changed cannabis use patterns and perceived levels of risk. Based on a nationwide survey, 668,157 Thais aged 12–15 years reported using cannabis in the past year (13.3 per 1,000 population), an increase of 3.5 times from 2016 (188,496 users), making it the most commonly used drug in 2019 (*Administrative Committee of Substance Academic Network, 2019*). Another survey in 2019 found that 86% of general people aged 15 years and over agreed with the policy on medical cannabis but only 31% agreed with the policy to allow its recreational use (*Centre for Addiction Studies, 2020*).

Amidst this background of extensive changes in policy, increased evidence of health effects, and the rapid escalation in the cannabis consumption either for medical or recreational purpose worldwide, it is important for policymakers and healthcare providers to understand how people consume cannabis for medical purposes, and how changes in cannabis legislation may impact patterns of consumption. In this study, we aimed to examine patterns and purposes of the consumption of medical cannabis and the consumers' perceptions and opinions towards the benefits and harms of cannabis and related policies.

## MATERIALS & METHODS

### Subjects and sampling method

This study used a respondent-driven sampling (RDS) method (*Heckathorn, 2014*) to recruit participants who were current consumers of medical cannabis. RDS is a probability-based sampling method where sampling procedure starts with a convenience sample of well-networked population members, referred to as seeds. After enrolment and completing

the interview, seeds receive a fixed number of coupons to recruit members from their social network (*Johnston, 2013*). The researchers keep track of who recruits whom and their numbers of social contacts. By recruiting long respondent chains, biases related with the initial convenience sample of seeds are detached from the final sample. The RDS method thus produces samples that are independent of the initial subjects from which sampling begins. It also combines the breadth of coverage of network-based methods with the statistical validity of standard probability sampling methods, making it possible to draw statistically valid samples of hard-to-reach population groups (*Heckathorn, 1997*). Although cannabis use for medical purposes was allowed in Thailand at the time of the survey, medical cannabis clinics had not yet opened in all MoPH hospitals, thus legal access was limited. Most consumers of cannabis were still considered as illegal consumers and a "hidden population". RDS was therefore justified as the method of choice for recruiting participants.

Consumers of medical cannabis in this study refer to individuals who had been using medicinal cannabis products (including raw plants) to treat or relieve their symptoms or health conditions within the past 12 months of the study. This definition does not imply that the cannabis products were indicated or prescribed by a health professional.

Four parallel recruitment sites were included: Chiang Mai, Khon Kaen, Bangkok, and Songkhla, representing the Northern, Northeastern, Central, and Southern regions of the country, respectively. Identical RDS procedures were used across the four sites. In each site, 3–4 seeds who were well connected to and trusted by the target population were identified through local contacts. In an attempt to recruit representative participants from various socio-demographic groups, the seeds were selected to include both males and females, three age-groups: young or middle adult (18–44 years), late adult (45–64) and elderly (>65), those who received medical cannabis from legal and illegal sources, and those who used it for different conditions (cancer and non-cancer patients). Participants were eligible for the study if they were a current consumer of medical cannabis, aged 18 years or over, and currently lived in one of the four study regions. Exclusion criteria included those who were intoxicated, cognitively or mentally impaired, or too ill to be interviewed; however, no subject was excluded due to any of these reasons. No more than three participants were allowed to be recruited from each recruiter. We aimed to recruit 120–125 participants from each site. This sample size was calculated assuming a design effect of 2 and was sufficiently powered to estimate an assumed medical cannabis use prevalence of 20% with an absolute precision of 10%.

## Measures

Face-to-face interviews using a structured questionnaire were used to collect data. The questionnaire contained four sections, including (1) demographic characteristics, (2) pattern of consumption (quantity, frequency, type, form, route of administration, indication for use, and source), (3) source of information and perception of benefits and harms of medical cannabis, and (4) opinion towards cannabis policies.

## Data collection procedure

We trained research assistants who were at least bachelor degree graduates and who had previous experience in data collection with people who use drugs in our other research projects. There were 2–3 research assistants at each site, making a total of 10 research assistants in the study. They were well trained in interviewing techniques and confidentiality protection. The seeds were contacted by phone and invited to participate in the study by the research assistants who explained the purposes, procedures, and data safeguards of the study. Interviews were done at the participant's home or other convenient places. After completing the questionnaire, each seed or successive participant was asked to invite three other participants to the study by giving them the research team's contact information. Those who were interested to participate in the study would contact us for the interview by themselves. This approach aimed to reduce masking because it gave respondents the opportunity to allow peers to decide for themselves whether or not they wanted to participate. However, this made us unable to calculate the response rate for the participants because we only knew the number who agreed to participate in the study—we did not know the number of people that each respondent invited.

Verbal informed consent was obtained from all participants. Data collection was conducted in private and participants were assured that any information disclosed would be treated in strict confidence. All data on participants were saved and analyzed anonymously. The study protocol, including informed consent procedures, was approved by the Research Ethics Committee of Faculty of Medicine, Prince of Songkla University (REC.62-205-18-1).

## Statistical analyses

We used RDS Analyst (*Handcock, Fellows & Gile*) to analyze the data. We pooled data of the four recruitment sites and normalized the RDS weights by site by multiplying the sampling weight of each participant by a further weighting, which was the mean of individual weights in each site divided by the sum of mean weights of all sites. The normalization was done in order to reduce the bias from different weightings among sites before pooling data of all sites for the analysis of data of all participants. Participant demographics and patterns of cannabis consumption and other variables were described using percentages and means or medians with 95% confidence interval (CI).

# RESULTS

## Sample characteristics and recruitment

We recruited 120–125 subjects from each site, making a total sample size of 485 altogether. The number of waves ranged from 3 (north-eastern and central regions) to 18 (southern region). There were more males, with the highest proportion seen in the north (66.2%, 95% CI [53.6–78.8]). The highest proportion of consumers was in the late adult age group. About one-third achieved a bachelor degree. The main occupation type was government officer (19.6%) followed by business owner (16.5%; Table 1).

## Patterns of cannabis consumption

Of all respondents, 22% had previously consumed cannabis for other purposes before their consumption for medical purposes. The duration of current consumption for medical

Assanangkornchai et al. (2022), *PeerJ*, DOI 10.7717/peerj.12809

**Table 1** Description of surveys and socio-demographic characteristics of medical cannabis users, % (95% confidence interval).

| Variable | North | Center | North-east | South | Total |
|---|---|---|---|---|---|
| Sample size (N) | 125 | 120 | 120 | 120 | 485 |
| Number of seeds (N) | 3 | 4 | 4 | 4 | 15 |
| Maximum waves (N) | 7 | 5 | 5 | 18 | 18 |
| Study period | 8 Oct–30 Nov 2019 | 18 Nov 2019–3 Feb 2020 | 11 Nov 2019–29 Feb 2020 | 8 Oct 2019–31 Jan 2020 | – |
| Data collection period | 8 weeks | 11 weeks | 16 weeks | 16 weeks | – |
| Sex: Male | 66.2 (53.6, 78.8) | 54.9 (42.8, 66.9) | 51.7 (39, 64.4) | 53.2 (41.6, 64.7) | 55.7 (49.4, 62.0) |
| Female | 33.8 (21.2, 46.4) | 45.1 (33.1, 57.2) | 48.3 (35.6, 61) | 46.8 (35.3, 58.4) | 44.3 (38, 50.6) |
| Age: 18–44 years | 16.3 (7.8, 24.8) | 45.9 (30.7, 61.1) | 18.4 (8.9, 28) | 16.9 (9.2, 24.6) | 22.8 (17.4, 28.1) |
| 45–65 years | 65.2 (53.6, 76.8) | 48.9 (34.9, 63) | 56.5 (44.8, 68.1) | 71.1 (61.3, 81.0) | 61.5 (55.7, 67.4) |
| ≥65 years | 18.5 (9.5, 27.6) | 5.1 (0.8, 9.4) | 25.1 (15.0, 35.1) | 11.9 (3.9, 20.0) | 15.7 (11.5, 20.0) |
| mean (min, max) | 56.1 (24.0, 96.0) | 48.5 (21.0, 73.0) | 57.2 (19.0, 87.0) | 53.8 (24.0, 87.0) | 53.9 (19.0, 96.0) |
| Education: Primary school/lower | 39.4 (25.5, 53.3) | 25.4 (13, 37.8) | 21.9 (10.6, 33.3) | 20.3 (10.9, 29.8) | 25.8 (19.6, 32.0) |
| Secondary school | 26.1 (14.7, 37.4) | 29.1 (18.5, 39.7) | 18.6 (9.4, 27.8) | 21.8 (11.6, 32.0) | 23.3 (18.2, 28.4) |
| Vocational college | 9.9 (3.5, 16.3) | 32.0 (18, 46.1) | 4.2 (1.2, 7.3) | 23.0 (13.0, 32.9) | 16.9 (11.7, 22.2) |
| Bachelor degree or higher | 24.7 (14.1, 35.2) | 13.4 (5.5, 21.4) | 55.2 (42.0, 68.4) | 34.9 (24.2, 45.6) | 34.1 (27.9, 40.3) |
| Occupation: Unemployed/Retired | 25.5 (11.0, 40.0) | 9.4 (1.4, 17.4) | 8.4 (0.7, 16.0) | 9.5 (2.6, 16.4) | 12.5 (7.5, 17.5) |
| Laborer | 8.6 (3.6, 13.5) | 16.4 (6.9, 26.0) | 6.2 (1.5, 10.9) | 3.8 (0, 8.0) | 7.9 (5.0, 10.9) |
| Farmer | 17.0 (8.5, 25.4) | 8.8 (1.2, 16.4) | 13.8 (4.1, 23.5) | 11 (3.6, 18.5) | 12.6 (8.5, 16.7) |
| Vendor | 13.7 (2.7, 24.6) | 15.9 (5.4, 26.4) | 6.2 (0, 12.5) | 17.4 (6.6, 28.2) | 13.2 (8.3, 18.1) |
| Private employee | 18.1 (3.9, 32.4) | 24.9 (13.2, 36.5) | 4.5 (0, 9.7) | 5.7 (1.1, 10.3) | 11.7 (7.2, 16.3) |
| Government officer | 3.9 (1.1, 6.8) | 3.6 (0, 7.3) | 32.4 (20.1, 44.6) | 28.6 (17.9, 32.2) | 19.6 (14.0, 25.1) |
| Business owner | 8.8 (1.7, 15.8) | 18 (8.5, 27.4) | 15.7 (3.6, 27.7) | 21.5 (10.7, 32.2) | 16.5 (11.5, 21.4) |
| Student, priest | 4.5 (0, 9.7) | 3.0 (1.4, 17.5) | 12.9 (4.1, 21.6) | 2.5 (0, 6.3) | 5.9 (2.9, 8.9) |

purposes was 10.5 months (range 7–828 days) on average. Two-third of the respondents (68.8%, 95% CI [61.4–76.2]) consumed it almost every day or many times a day every day, with 72.6% (95% CI [64.9–80.2]) reporting the same pattern of consumption since the beginning of medical use. Most respondents (79.1%, 95% CI [69.3–89.0]) reported their conditions were ameliorated after they started using cannabis. An oral intake of crude oil extract (unidentified tetrahydrocannabinol (THC) or cannabidiol (CBD) content) was the most common form of consumption reported by 84.7% (95% CI [78.9–90.5]) of the respondents. Other forms included raw plants (flowers, leaves or whole plants with roots and stems; 9.2% (95% CI [4.1–14.2])) and topical skin products (massage oil, cream, spray, soap; 5.0% (95% CI [2.2–7.8])).

## Conditions treated with medical cannabis

The three most common conditions cannabis was used for treatment included malignant or non-malignant tumors (23.3%, 95% CI [16.1–30.4]); neuro-psychiatric disorders (22.8%, 95% CI [17.5–28.0]) and musculoskeletal symptoms, such as pains, spasm, rigidity or weakness (21.6%, 95% CI [16.7–26.6]). Other conditions were diverse, for instance: diabetes mellitus, hypercholesterolemia, hypertension, asthma, HIV-AIDS, herpes zoster, herpes simplex, psoriasis and vitreous degeneration (Table 2).

Based on the disease groups indicated by the MoPH (5), proportions of the respondents reporting the consumption for diseases in Groups A (strong evidence for the benefits of cannabis, 21.5%), B (some evidence of benefits, 20.6%) and C (not enough evidence at present, 21.7%) were similar. Nevertheless, the highest proportion occurred in the "other" group, which was conditions not indicated for medical cannabis treatment by the MoPH (36.3%; Table 2).

## Sources of medical cannabis products

The majority of the respondents (74.0%, 95% CI [63.7–84.3]) acquired medical cannabis products from the illegal sources, including underground traders (54.5%, 95% CI [40.8–68.3]), friends and relatives (12.2%, 95% CI [6.2–18.3]), not-for-profit provider groups (5.2%, 95% CI [0.5–10.9]), and home or clandestine growers or producers (2.9%, 95% CI [0.6–5.3], Table 2). However, 26% accessed to the legal sources, for example modern (0.4%, 95% CI [0–1.0]) and Thai traditional medicine doctors (7.2%, 95% CI [0–17.7]) in medical cannabis clinics of MoPH hospitals, medical doctors in private practices (12.8%, 95% CI [0–25.8]), and folk doctors who were certified for using cannabis in their practice (4.6%, 95% CI [0–9.2]).

## Sources of information about medical cannabis

The main source where respondents obtained information about medical cannabis was from friends and relatives (78.3%), followed by social media (Facebook, Line Group; 32.9%) and disease-specific user network or advocacy groups, for example, friends of cancer patients network, mothers of epileptic children network and medical cannabis advocacy group. Only 15.4% reported receiving information from healthcare providers or government organizations (Table 2).

Assanangkornchai et al. (2022), *PeerJ*, DOI 10.7717/peerj.12809

**Table 2  Conditions for use and sources of medical cannabis.**

| Variable | North | Center | North-east | South | Total |
|---|---|---|---|---|---|
| **Condition for use**[*] | | | | | |
| Cancers | 30.6 (9.7, 51.6) | 2.2 (0, 4.6) | 22.8 (13.3, 32.3) | 31.9 (15.7, 48.1) | 23.3 (16.1, 30.4) |
| Diseases or symptoms of the musculoskeletal system | 25.3 (11.2, 39.4) | 28.6 (17.9, 39.3) | 14.6 (6.9, 22.4) | 21.2 (11.0, 31.5) | 21.6 (16.7, 26.6) |
| Neuro-psychiatric symptoms or disorders | 8.2 (3.0, 13.5) | 32.5 (20.9, 44.0) | 27.3 (15.8, 38.7) | 22.3 (12.0, 32.5) | 22.8 (17.5, 28.0) |
| Non-communicable diseases | 26.6 (13.2, 40.0) | 32.6 (21.6, 43.7) | 21.7 (12.2, 31.3) | 20.5 (8.7, 32.3) | 24.5 (19.1, 29.9) |
| Others | 9.2 (0,18.5) | 4.2 (1.9, 6.4) | 13.6 (3.6, 23.6) | 4.1 (0, 8.4) | 7.9 (4.0, 11.7) |
| **Condition for use**[**] | | | | | |
| A. With strong evidence of benefits | 21.5 (7.8, 35.2) | 27.2 (16.8, 37.5) | 13.8 (5.8, 21.7) | 24.8 (13.8, 35.8) | 21.5 (16.3, 26.6) |
| B. With some evidence of benefits | 7.9 (2.8, 13.0) | 32.5 (21.5, 43.4) | 23.9 (13.3, 34.5) | 18.7 (9.3, 28.1) | 20.6 (15.4, 25.8) |
| C. With Not enough evidence at present | 26.1 (6.9, 45.4) | 2.2 (−0.1, 4.5) | 20.4 (11.5, 29.3) | 31.9 (15.8, 48.0) | 21.7 (15.3, 28.1) |
| D. Others | 44.5 (28.6, 60.3) | 38.2 (26.6, 49.7) | 41.9 (28.9, 54.9) | 24.6 (13.2, 36.0) | 36.3 (29.8, 42.8) |
| **Source of cannabis**[***] | | | | | |
| Legal source | 24.3 (11.4, 37.3) | 0 | 70.2 (51.0,89.5) | 2.5 (0, 5.9) | 26.0 (15.7, 36.3) |
| Illegal source | 75.7 (62.7, 88.7) | 100 | 29.8 (10.5, 49.0) | 97.5 (94.1, 100.8) | 74.0 (63.7, 84.3) |
| **Source of information** | | | | | |
| User network/advocacy group | 7.7 (2.8, 12.7) | 4.9 (0, 10.6) | 40.3 (19.9, 60.6) | 14.1 (6.9, 21.3) | 18.4 (12.3, 24.5) |
| Healthcare provider/government organisation | 0.3 (0, 0.9) | 0 | 49 (29.4, 68.6) | 4.8 (0.3, 9.3) | 15.4 (6.4, 24.4) |
| Sale website | 4.1 (0, 13.4) | 0 | 44.1 (22, 66.3) | 3.9 (0.5, 7.2) | 14.5 (3.6, 25.5) |
| Social media | 16.8 (5.7, 28) | 53.4 (35.9, 70.9) | 55.1 (42, 68.2) | 10.6 (4.6, 16.7) | 32.9 (25.6, 40.1) |
| Friends, relatives | 71.9 (61.6, 82.3) | 92 (81.4, 102.5) | 77.6 (67, 88.2) | 74.5 (65.6, 83.4) | 78.3 (72.8, 83.8) |
| Sellers (on site) | 12.2 (4.9, 19.4) | 0 | 2 (0, 4.8) | 3.1 (0.1, 6.1) | 4 (1.6, 6.5) |

**Notes.**

[*]Conditions for use of medical cannabis: Cancers included cancers of breast, prostate gland, lymph nodes, liver, lung, intestine, ovary, ureter and bladder; Diseases or symptoms of musculoskeletal system included muscle or joint pains, spasm, rigidity or weakness; Neuro-psychiatric symptoms or disorders included stress, depression, anxiety, bipolar affective disorders, insomnia, stroke, epilepsy, multiple sclerosis, dementia and Parkinson disease; Non-communicable diseases included diabetes mellitus, hypercholesterolemia, hypertension and gout; Other conditions included asthma, chronic lung disease, anemia, liver cirrhosis, HIV-AIDS, herpes zoster, herpes simplex, psoriasis, vitreous degeneration, cataract and low appetite.

[**]Conditions for which medical cannabis is classified by the Ministry of Public Health (MoPH) based on supporting evidence: A. Conditions with strong evidence of benefits from medical cannabis, *i.e.*, chemotherapy induced nausea and vomiting, intractable epilepsy, spasticity in patients with multiple sclerosis and neuropathic pain, B. Conditions with some evidence of benefits, *i.e.*, patients in palliative care, patients with end-stage cancer, Parkinson disease, Alzheimer disease, generalized anxiety disorder and other demyelinating diseases, C. Conditions which may be benefited from treatment with cannabis should there be more evidence in the future, *e.g.*, cancers of some organs, and D. Other conditions, which are yet supported by MoPH as having evidence of benefit from medical cannabis treatment (Ministry of Public Health, 2020).

[***]Source of cannabis: (1) Legal sources included medical cannabis clinics (modern and traditional medicine) in MoPH hospitals and private clinics where providers have been certified by the MoPH, and (2) Illegal sources included illegal traders, not-for-profit provider groups and clandestine growers or producers.

### Perceptions of benefits and harms of cannabis

Most (65.9% to 89.9%) perceived that cannabis could be used to treat all conditions suggested by the MoPH. About 90% of respondents perceived that cannabis could treat cancers while almost 100% said it helped with insomnia and 80% said it increased appetite and decreased weight loss in HIV/AIDS patients and also improved post-traumatic stress disorder (PTSD) symptoms. Furthermore, more than half of the respondents believed that cannabis could treat substance dependence, brain tumors, and chronic cough (Table 3).

Less than half of the participants perceived that cannabis could cause some adverse conditions such as palpitations, panic, dementia, memory impairment or amotivational syndrome, schizophrenic-like psychosis, abnormal locomotor movements which increased accidental risk and hallucinations.

### Opinions towards cannabis-related policies and measures in Thailand

Most respondents agreed or strongly agreed with the policies regarding permission to use cannabis for medical purposes (95.1%, 95% CI [92.0–98.2]), for the legal sale of medical cannabis products (95.9%, 95% CI [93.7–98.2]), and for people to grow cannabis for medical use (94.2%, 95% CI [91.8–96.5]). However, only two-thirds agreed or strongly agreed with policies concerning the sales of cannabis (65.3%, 95% CI [56.9–73.7]) and home-growing cannabis for recreational purposes (61.3%, 95% CI [52.7–69.9]). Additionally, 80% (95% CI [74.2–85.9]) of participants agreed or strongly agreed that the cannabis industry would benefit the economy. When asked about their preferred legal status of cannabis, most respondents stated that the legal control of cannabis should be at the same level as alcohol (75.0%, 95% CI [69.1–81]) or tobacco (75.8%, 95% CI [69.9–81.7]), however 21.6% (95% CI [15.9–27.2]) viewed that cannabis should remain under the narcotics control law as an addictive substance and be controlled at the same level as other substances of abuse such as heroin and methamphetamine.

## DISCUSSION

This study provides some insights into medical cannabis consumption in Thailand from the consumers' perspectives. It reflects the situation in late 2019 to early 2020, almost one year since new legislation was passed concerning medical cannabis use. There is little in the study findings to suggest large changes in the scenery of medical cannabis consumption in Thailand since it was legalized in February 2019. Most consumers still obtained medical cannabis from illegal sources and perceived a high level of effectiveness in treating a wide range of health conditions. This situation is consistent with that found in other countries, for example Canada, the USA and Australia early after the introduction of legal access pathways (Sexton et al., 2016; Lucas & Walsh, 2017; Lintzeris et al., 2020).

The finding that only 26% of the consumers of medical cannabis obtained the products legally can be explained as follows. First, data collection occurred when MoPH medical cannabis clinics were available in only a few provinces. Furthermore, the recommended indications for the use of medical cannabis by the MoPH are very limited (Department of Medical Services, 2020) and the attitudes of some health professionals, such as psychiatrists

Assanangkornchai et al. (2022), *PeerJ*, DOI 10.7717/peerj.12809

**Table 3** Proportions of respondents who perceived the benefits and harms of medical cannabis.

| Perceptions of benefits of medical cannabis | % (95% CI) | Perceptions of harms of medical cannabis | % (95% CI) |
| --- | --- | --- | --- |
| Treatment of chronic pain in adults | 83.6 (78.2, 88.9) | Palpitations | 42.3 (35.8, 48.8) |
| An antiemesis for patients who receive chemotherapy | 65.9 (59.3, 72.5) | Panic symptoms | 35.3 (28.1, 42.6) |
| Treatment of intractable epilepsy in children | 73.6 (67.6, 79.6) | Dementia, memory impairment, amotivation | 32.4 (26.0, 38.7) |
| Treatment of spasticity in multiple sclerosis | 73.8 (67.5, 80.1) | Schizophrenia-like psychotic symptoms | 31.9 (26.0, 37.8) |
| Treatment of Parkinson's disease | 79.1 (73.2, 85.1) | Severe dry mouth | 40.9 (34.5, 47.3) |
| Treatment of Alzheimer's disease | 73.4 (67.5, 79.2) | Slow reaction time, abnormal sensory-motor function | 33.4 (27.3, 39.6) |
| Treatment of generalized anxiety disorder | 82.6 (77.7, 87.6) | Hallucinations | 34.4 (28.0, 40.7) |
| Treatment of cancers | 89.9 (86.0, 93.8) | Acute hypotension | 24.4 (18.6, 30.3) |
| Improvement of insomnia | 99.1 (97.9,100) | Decreased sperm count, infertility | 18.4 (13.1, 23.7) |
| Increased appetite in HIV/AIDS patients | 77.0 (71.0, 82.9) | Ataxia, uncontrollable body coordination | 37.8 (31.8, 43.8) |
| Improved PTSD symptoms | 78.6 (72.9, 84.3) | Blurred vision | 36.2 (30.1, 42.4) |
| Treatment of substance dependence | 51.9 (44.0, 59.8) | Hepatitis | 10.3 (6.8, 13.9) |
| Treatment of brain tumour | 50.1 (41.5, 58.8) | | |
| Decreased severity of chronic cough | 62.6 (55.7, 69.5) | | |

**Notes.**

PTSD, Post-traumatic stress disorder.

and pharmacists, are non-supportive and skeptical, making healthcare providers reluctant to prescribe them. Second, most consumers had been consuming cannabis either for medical or recreational purposes long before the legal access was available (10.5 months on average). They had obtained it from illegal sources and continued acquiring it from the same sources as they were easily accessible and affordable. Additionally, those non-government providers have formed networks of providers and advocates who promote its medical use through word of mouth or social media. Some may provide their products for free or at a low cost and send the products to the consumers' homes by mail, making it very convenient for consumers. These findings all reflect the accessibility situation of medical cannabis at the time of the study, which was limited in terms of indications for use and the attitudes of providers and staff at service clinics. As medical cannabis is a national priority policy, strategies to increase patients' awareness and accessibility to legal provider sources, which are certainly safer than the illicit sources, are necessitated.

Our findings raise concerns about the illicit supplies of medical cannabis products, of which the cannabinoid content (*e.g.*, THC and CBD) is generally not known and the production process cannot be qualified. Consumption of cannabis, especially if it is high in THC, is strongly associated with several adverse health effects such as cardiovascular diseases and mental health problems (*Subramaniam et al., 2019*; *Latif & Garg, 2020*). Medical cannabis obtained from illicit sources can be similar in form to recreational cannabis or even have higher potency if the crude oil extract is used, which was the most common form consumed by our study participants. A paper in the USA reports that the highest THC level available for researchers is 12.4% while THC levels sold in the market averages 18.7% and some strains even exceed 35% (*Stith & Vigil, 2016*). There has been no research on the cannabinoid content of cannabis products in Thailand but there is some anecdotal evidence that the products are contaminated with a wide variety of insecticides and fungicides containing toxins as well as butane solvent which is used in the extraction process. An increasing number of cases with cannabis intoxication or other adverse events reported from hospitals in the 2–3 years after medical cannabis became popular (*Ramathibodi Poison Center, 2020*) can be an indicator of the widespread consumption of these low-quality products and should be a public health concern.

Currently, over 700 MoPH medical cannabis clinics are operated nationwide and the products prescribed are under the SAS (*Ministry of Public Health, 2020*). It seems likely that patients who fit the MoPH guidelines will turn from illicit sources to MoPH clinics, especially those with low incomes. This may increase access to legal and certified suppliers and in the long run, will hopefully decrease illicit trade and production.

We also found that most (66–90%) consumers saw only the positive side of cannabis and believed that it could treat or even cure any diseases, with 90% perceiving that it could treat cancers. Only a few consumers were aware that cannabis could cause some adverse effects such as palpitations, psychotic symptoms, and cognitive impairment. Furthermore, most consumers received information about medical cannabis from non-formal sources, especially friends, relatives, social media, and advocacy groups while only 15% reported obtaining information from governmental healthcare sources. These results are consistent with a survey among the general population in November 2019 in Thailand, which found

a high percentage of respondents perceived cannabis could cure cancers and revealed television, social media, and word of mouth as the most common sources of information (*Centre for Addiction Studies, 2020*). Another study in 2019 in Thailand found that common social discourses towards cannabis included "cannabis is a medical hero" and "cannabis is a Thai folk wisdom" (*Runkasem, 2020*). The reasons to support these findings may be because most media provide more positive aspects of cannabis or the consumers choose to perceive only the positive side of the information to support their personal beliefs. This social atmosphere could explain why most consumers in our study perceived more benefits than harms of cannabis, regardless of the evidence.

We found that the most common conditions for consumption of cannabis, namely cancer, pains of the musculoskeletal system, and mental illnesses, were similar to those found in other studies (*Webb & Webb, 2014*; *Sexton et al., 2016*; *Lintzeris et al., 2020*). The highest percentage (36%) of our participants used cannabis for conditions that had no evidence for its efficacy, based on the MoPH classification (*Ministry of Public Health, 2020*). These findings can be explained by the fact that most consumers perceived more benefits than harms from using cannabis and believed it can treat any disease. Additionally, they could obtain cannabis from illegal suppliers who could prescribe it for any condition without the need to follow the MoPH recommendations. This thus made them able to use cannabis for any condition, even one not included in the recommendations. These findings may reflect that the conditions recommended by the MoPH may be too limited as they are developed based on available supporting evidence and do not adequately fulfil cannabis consumers' needs, especially for cancer patients whose condition is still listed in category 3 which needs more evidence to support its efficacy. However, for cancer patients, particularly those in the late stage where other treatment is not available or affordable, medical cannabis may be their only available option and satisfies their need regardless of any supporting evidence. Clinical guidelines usually need continuous updating when more evidence becomes available. Patients' preferences and needs should also be taken into account when updating the guidelines and this work should be one of the MoPH priority actions to increase access to a legal supply and safe consumption of medical cannabis.

Most participants in our study supported the policy of unlocking cannabis for medical and not for recreational purposes (94–96% vs. 61–65%). Some respondents cited that evidence of the benefits of medical cannabis exists from other countries and Thai traditional medicine and that if it was controlled at the same level as alcohol or tobacco, it would create more revenue for the government. However, some respondents expressed concerns over potentially increasing cannabis use by adolescents if it was liberalized and some said a good system was needed for the safe use of cannabis. This finding is consistent with other studies in Thailand and other countries (*McGinty et al., 2017*; *NIDA Poll, 2019*; *Resko et al., 2019*; *Centre for Addiction Studies, 2020*). It confirms that Thai people generally accept a medical cannabis policy but guidelines for its monitoring and control should be clear and strong with effective public communication to make people understand and use it with caution.

This study has some limitations. Although we used an RDS method, which resulted in some seeds having very long chains and thus making recruits independent from each

other, and we used seeds living in 2–3 provinces in each region, our respondents tended to concentrate in the provinces where the main data collection centers were situated. Compared to the general Thai population (*National Statistics Office & Ministry of Digital Economy and Society, 2020*) and consumers of cannabis identified in a general population survey (*Centre for Addiction Studies, 2020*), people of middle or higher socioeconomic status, *e.g.*, bachelor degree graduates, government officers and business owners were over-represented among our study participants. This occurred because most of our RDS seeds who were known medical cannabis consumers were of these socio-economic groups and therefore our sample may not represent medical cannabis consumers in the whole country. In addition, young consumers were usually anonymously engaged in virtual networks and tended to buy cannabis products online rather than through a physical network, so it was difficult to identify this group of new generation consumers. Most of the respondents were in the middle and older age groups who were physically connected; thus, they did not well represent consumers of the new generation. Lastly, our data collection started when the legal supply of cannabis had just been started in Thailand. Therefore, we could only recruit a few consumers who obtained medical cannabis from legal sources and could obtain only limited information regarding the effects of consumption and satisfaction towards medical cannabis services in MoPH clinics. These limitations may affect generalizability of the results to the general medical cannabis consumers of the country who may be of different socio-demographic and socio-economic groups. However, they did not threaten internal validity of the study.

## CONCLUSION

We described patterns and purposes of medical cannabis consumption in Thailand and perceived benefits and harm according to consumers during the first year after the major regulatory transition. Consumers reported various patterns and indications of consumption that were not supported by scientific evidence, but had positive perception of the results of consumption. There remains a lack of information on how consumers will transition to legal sources of cannabis after medicinal-grade cannabinoid products are added to the list of essential medicines subsidized by the Thai government. Given that the majority of our participants reported that they obtained cannabis from illegal sources, we believe that there is an urgent need to facilitate access to high-quality legal products, revise prescription indications with updated scientific evidence, and to provide effective public communication to protect the public's health. The findings of this study highlighted ongoing policy challenges and may be of interest to other countries in the region considering similar changes.

## ACKNOWLEDGEMENTS

The authors wish to thank Edward McNeil for his advice on statistical analysis and English proof reading of the manuscript.

### Funding

This work was supported by the Health System Research Institute through the National Health Foundation (CNB 62001). The funders had no role in study design, data collection and analysis, decision to publish, or preparation of the manuscript.

### Grant Disclosures

The following grant information was disclosed by the authors:
Health System Research Institute through the National Health Foundation: CNB 62001.

### Competing Interests

The authors declare there are no competing interests.

### Author Contributions

- Sawitri Assanangkornchai conceived and designed the experiments, performed the experiments, prepared figures and/or tables, authored or reviewed drafts of the paper, and approved the final draft.
- Kanittha Thaikla conceived and designed the experiments, performed the experiments, analyzed the data, prepared figures and/or tables, authored or reviewed drafts of the paper, and approved the final draft.
- Muhammadfahmee Talek and Darika Saingam performed the experiments, authored or reviewed drafts of the paper, and approved the final draft.

### Human Ethics

The following information was supplied relating to ethical approvals (i.e., approving body and any reference numbers):
The study protocol, including informed consent procedures, was approved by the Research Ethics Committee of Faculty of Medicine, Prince of Songkla University (REC.62-205-18-1).

### Field Study Permissions

The following information was supplied relating to field study approvals (i.e., approving body and any reference numbers):
Field interviews were approved by the Research Ethics Committee of Faculty of Medicine, Prince of Songkla University (REC.62-205-18-1).

### Data Availability

The raw data are available in the Supplementary File.

### Supplemental Information

Supplemental information for this article can be found online at http://dx.doi.org/10.7717/peerj.12809#supplemental-information.

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
