# Peer review of "Medical cannabis use in Thailand after its legalization: a respondent-driven sample survey"

_PeerJ, doi:10.7717/peerj.12809_

## Round 0.1 · original submission · Minor Revisions

Dear Dr. Assanangkornchai,

Thank you for your submission to PeerJ. The manuscript has been reviewed and we believe that it has merit but cannot be published in its current form as it requires a number of revisions.

Please address all the issues raised by the reviewers below as well as in the attached file. Although not a hard deadline, please submit your revised manuscript within the next 35 days.

Sincerely,

Agricola Odoi
Academic Editor, PeerJ
====================
Reviewer 1#
Please go through the attached file and reply to the reviewer comments

Reviewer 2#
Additional comments
This is a well-written and well-designed study that contributes important information regarding the legalization of medical cannabis. While this journal does not evaluate impact, I do want to comment on this. As more and more jurisdictions move towards various forms of decriminalization/legalization it is important that policymakers have access to good data that helps increase understanding of how people make decisions regarding their cannabis consumption. This piece adds to this data in a positive way. I would be happy for the article to be published as is, but I do have some suggestions that the editor and authors might wish to consider. All of these are monitor and are offered in the spirit of collegiality.

Intro:
The framing of the issue and intro is well done. Provides good information to the international reader. I think lines 318-322 (the history and context relating to legislation) would go better in the intro. I was looking for this information in the intro and was happy to see it at the end, but I moving it to the intro will improve the piece.

I appreciate the justification for this study is not based on “gaps” but on the importance of the information for policy development.

Methods:
Nice description of RDS and justification for this approach

Line 157 - “drug users” may be accurate but this language is being critiqued for contributing to stigma. Some authors/researchers use PWUD (people who use drugs). “Consumers of cannabis” would also be fine. I recommend changing “users” to consumers throughout. I think this will hold up better over time and make your work more likely to be cited, especially by a North American audience.

Sample characteristics: For an international audience it would be helpful to know how your participant demographic compares to the population in question. E.g. Is the rate of the type of employment (government officer) higher than expected? Does the age of your participants skew in one direction or another?

Line 185 - I suggest: “Of all respondents, 22% had previously used cannabis…” I think "previously" rather than"ever" improves readability

Patterns of cannabis use - Again might want to say cannabis consumption rather than use. It is interesting that the majority used oils over raw flowers!

Also interesting is the low rate of information from physicians and healthcare providers vs Facebook and social media. Important implications for policy.

Opinions towards cannabis-related policies
This is a minor issue but I would rephrase Lines 249 - 253 so that the majority opinions come first (as you have elsewhere). Without a close reading, it first seems that the majority want legalization at the same level as narcotics when in fact this is the smallest group.

Overall I am very pleased with this piece. Thank you for the opportunity to review your work. I wish you continued success in your research.

Reviewer 1 ·

Basic reporting

no comment

Experimental design

no comment

Validity of the findings

no comment

Annotated reviews are not available for download in order to protect the identity of reviewers who chose to remain anonymous.

·

Basic reporting

Well written and well sourced

Experimental design

Well justified and carried out

Validity of the findings

No comment

Additional comments

This is a well-written and well-designed study that contributes important information regarding the legalization of medical cannabis. While this journal does not evaluate impact, I do want to comment on this. As more and more jurisdictions move towards various forms of decriminalization/legalization it is important that policy makers have access to good data that helps increase understanding of how people make decisions regarding their cannabis consumption. This piece adds to this data in a positive way. I would be happy for the article to be published as is, but I do have some suggestions that the editor and authors might wish to consider. All of these are monitor and are offered in the spirit of collegiality.

Intro:
Framing of the issue and intro are well done. Provides good information to the international reader. I think lines 318-322 (the history and context elating to legislation) would go better in the intro. I was looking for this information in the intro and was happy to see it at the end, but I moving it to the intro will improve the piece.

I appreciate the justification for this study is not based on “gaps” but on the importance of the information for policy development.

Methods:
Nice description of RDS and justification fro this approach

Line 157 - “drug users” may be accurate but this language is being critiqued for contributing to stigma. Some authors/researchers use PWUD (people who use drugs). “Consumers of cannabis” would also be fine. I recommend changing “users” to consumers throughout. I think this will hold up better over time and make your work more likely to be cited, especially by a North American audience.

Sample characteristics: For an international audience it would be helpful to know how your participant demographic compares to the population in question. E.g. Is the rate of type of employment (government officer) higher than expected? Does the age of your participants skew in one direction or another?

Line 185 - I suggest: “Of all respondents, 22% had previous used cannabis…” I think "previously" rather than"ever" improves readability

Patterns of cannabis use - Again might want to say cannabis consumption rather than use. Interesting that the majority used oils over raw flower!

Also interesting the low rate of information from physicians and healthcare providers vs Facebook and social media. Important implications for policy.

Opinions towards cannabis-related policies
This is a minor issue but I would rephrase Lines 249 - 253 so that the majority opinions come first (as you have elsewhere). Without a close reading it first seems that the majority want legalization at the same level as narcotics when in fact this is the smallest group.

Overall I am very pleased with this piece. Thank you for the opportunity to review your work. I wish you continued success in your research.

---

## Round 0.2 · Minor Revisions

Thanks for addressing the issues raised by both reviewers. I think the manuscript has merit and has been improved. However, I noticed that you did not address the comments provided in the attached file---it seems you did not see the attachment. Please address all the comments and suggested edits in the attached file and re-submit the manuscript.

Thank you and I look forward to reviewing your revised manuscript.

Reviewer 1 ·

Basic reporting

Title: The title identify and cover the main content of the manuscript .

Abstract: adequately state the research objectives, methods, results, conclusions and recommendations.

Introduction: reflect the important content of the manuscript.

Experimental design

The materials, equipment and research procedures described with sufficient detail.

Validity of the findings

Results: Statistical analysis were suitable for the research methodology, and results presented in tables/figures were correct.

Additional comments

I satisfied with all aspects of the paper ; in other words, the manuscript is publishable.

·

Basic reporting

Well done

Experimental design

Well done

Validity of the findings

Well done

Additional comments

Well done

---

## Round 0.3 · accepted · Accept

Although the manuscript has scientific merit and is fit for publication, there are still a number of grammatical errors in the document. Please ensure that you carefully read and correct these errors during the proofing stage of the manuscript.

Again, congratulations and thanks for choosing PeerJ.